# Determination of Tropical Belt Widening Using Multiple GNSS Radio Occultation Measurements

**Mohamed Darrag[1,2], Shuanggen Jin[1,3,4]\*, Andrés Calabia[1] and Aalaa Samy[2]**

[1]School of Remote Sensing and Geomatics Engineering, Nanjing University of Information Science and Technology, Nanjing 210044, China

[2]National Research Institute of Astronomy and Geophysics-NRIAG, 11421- Helwan, Cairo, Egypt

[3]Shanghai Astronomical Observatory, Chinese Academy of Science, Shanghai, China

[4]School of Surveying and Land Information Engineering, Henan Polytechnic University, Jiaozuo 454000, China

*Corresponding author: sgjin@nuist.edu.cn; sg.jin@yahoo.com

## Abstract

In the last decades, several studies reported the tropics expansion but the rates of expansion are widely different. In this paper, data of 12 global navigation satellite systems radio occultation (GNSS-RO) missions from June 2001 to November 2020 with high resolution were used to investigate the possible widening of the tropical belt along with the probable drivers and impacts in both hemispheres. Applying both lapse rate tropopause (LRT) and cold point tropopause (CPT) definitions, the global tropopause height shows an increase of approximately 36 m/decade and 60 m/decade, respectively. The tropical edge latitudes (TELs) are estimated based on two tropopause height metrics, subjective and objective methods. Applying both metrics, the determined TELs using GNSS have expansive behavior in northern hemisphere (NH) while in southern hemisphere (SH) there are no significant trends. In case of ECMWF Reanalysis v5 (ERA5) there are no considerable trends in both hemispheres. For Atmospheric Infrared Sounder (AIRS), there is expansion in NH and observed contraction in SH. The variability of tropopause parameters (temperature and height) is maximum around the TEL locations in both hemispheres. Moreover, the spatial and temporal patterns of total column ozone (TCO) have good agreement with the TELs positions estimated using GNSS LRT height. Carbon dioxide ($CO_2$) and Methane ($CH_4$), the most important greenhouse gases (GHGs) and the main drivers of global warming, have spatial modes in the NH that are located more poleward than that in the SH. Both surface temperature and precipitation have strong correlation with GNSS LRT height. The surface temperature spatial pattern broadly agrees with the GNSS TEL positions. In contrast, Standardized Precipitation Evapotranspiration Index (SPEI) has no direct connection with the TEL behavior. The results illustrate that the tropics widening rates are different from data set to another and from metric to another. In addition, TEL behavior in NH is different from that in SH. Furthermore, the variability of meteorological parameters agrees with GNSS TEL results more than with that of other data sets.

**Keywords:** GNSS-RO, Tropopause, Tropical belt, Climate change.

## 1. Introduction

Several studies have reported a widening of the tropics in observations, model simulations and reanalyses. This expansion may lead to profound changes in the global climate system, even a minor expansion of the tropical belt would have significant implications because the shift of the jet streams and subtropical dry zones toward poles have direct effects on weather and precipitation patterns. The widening of the tropical belt is largely considered to be a response to global warming

caused by increased GHGs concentrations (Davis and Rosenlof, 2012; Davis and Birner, 2013; Staten et al., 2018; Grise et al., 2019; Watt-Meyer et al., 2019; Meng et al., 2021; Pisoft et al., 2021). The reported widening rates, in most of previous studies, range from 0.25° to 3.0° latitude/decade and their statistical significance vary by large amount based on the metrics used to estimate the tropical edge latitude (TEL) as well as the data sets utilized for its derivation. In addition, the used metrics may respond in different ways to the force driving the widening because of their differing physics (Davis and Rosenlof, 2012). Hudson et al. (2006), based on atmospheric ozone concentrations, reported that northern hemisphere (NH) occupied by the tropical region grew at a rate of 1°/per decade. Using independent set of satellite-based microwave observations of atmospheric temperature, Fu et al. (2006) inferred tropical belt widening for the period 1979–2005. They estimated a net widening of about 2° latitude. Based on radiosonde (RS) and reanalysis data Seidel and Randel (2007) reported an expansion of 5 to 8 degrees latitude during the period from 1979 to 2005. In addition, Hu and Fu (2007) found a widening of the tropical Hadley circulation system, and estimated its magnitude as 2° to 4.5° latitude during period from 1979 to 2005. Ao and Hajj (2013) used GPS RO data over the period 2002 to 2011 and analyzed it to examine the possible expansion of the tropical belt due to climate change. Their analysis showed a statistically significant widening trend of 1°/decade in the NH while in Southern Hemisphere (SH) no significant trend was found.

In astronomy and cartography, the edges of the tropical belt are the Tropics of Cancer and Capricorn, at latitudes of ~23.5° north and south, where the Sun is directly overhead at solstice. They are determined by the tilt of the Earth's axis of rotation relative to the planet's orbital plane, and their location varies slowly, predictably and very slightly by about 2.5° latitude over 40,000 years (Gnanadesikan and Stouffer, 2006). In climatology, tropics edges vary seasonally, interannually, and in response to climate forcing. They move poleward in the summer and equatorward in the winter (Davis and Birner, 2013). There are several indicators that define the boundaries of the tropical belt. Generally, three main classes of metrics are employed to estimate the tropical belt borders: circulation-based metrics (e.g., based on the Hadley cells and the subtropical jets), temperature-based metrics (e.g., based on tropopause characteristics), and surface climate metrics (e.g., based on precipitation and surface winds) (Waliser et al., 1999). The common metrics used for TEL determination are discussed in details in Staten et al. (2018) and Adam et al. (2018). TELs estimated applying different metrics not all necessarily yield the same location. Their positions vary by much larger amounts and much more rapidly and unpredictably than the astronomically defined tropics (Lee and Kim, 2003).

Study of tropical belt widening is a challenging task due to the complexity and dynamics of the Earth's atmospheric system and the data limitations. These limitations are the low spatial resolution of RS data as it only covers land and its distribution is not symmetrical in both hemispheres. For the satellite remote sensing technologies and model analyses both suffer from low vertical resolution. Furthermore, reanalyses trends can be biased to reflect changes in both the quality as well as the quantity of the underlying data and the expansion rates computed from different reanalyses were considerably different (Schmidt et al., 2004; Ao and Hajj, 2013). Nowadays, Global navigation satellite systems (GNSS) have provided an exceptional opportunity to retrieve land surface and atmospheric parameters globally (e.g., Jin and Park, 2006; Jin and

Zhang, 2016; Wu and Jin, 2014; Jin et al., 2011, 2017), particularly space-borne GNSS Radio
Occultation (GNSS-RO) because GNSS-RO has long-term stability and works in all-weather-
conditions, which make it a powerful tool for studying climate variability. Since GNSS-RO has
uniform global coverage, it covers all locations even at the polar regions and oceans, which are
blind zones of other detection systems such as RS and radar. Moreover, GNSS-RO observations
vertically finer resolved than any of the existing satellite temperature measurements available for
the upper-troposphere lower-stratosphere (UTLS) thus GNSS-RO is well suited for this challenge.
Moreover, it is a key component for a broad range of other studies, including equatorial waves,
Kelvin waves, gravity waves, Rossby and mixed Rossby–gravity waves, and thermal tides (Bai et
al., 2020; Scherllin-Pirscher et al., 2021). A number of studies confirmed the feasibility and
excellent eligibility of GNSS-RO measurements for monitoring the atmosphere and for climate
change detection (Foelsche et al., 2009; Steiner et al., 2011).
Nowadays, GNSS-RO is a valuable remote sounding technique for the atmosphere. During
the GNSS-RO event, the GNSS satellite transmit signals that are received onboard a low earth
orbiting (LEO) satellite. Due to the atmospheric refractivity, these signals suffer time delay and
bending. The atmosphere excess propagation (AEP) is the main observable, and can be calculated
with millimeters accuracy, providing high quality and global observations (Wickert et al., 2001a).
For instance, the AEP estimate is the base to extract the profiles of bending angle, refractivity, and
temperature (Wickert et al., 2004; Xia et al., 2017). The GNSS-RO technique was firstly performed
within the US GPS/METeorology experiment for the period from 1995 to 1997 (Kursinski et al.,
1997). Also, it is continuously applied aboard various LEO satellite missions since 2001. "These
missions are Challenging Mini-satellite Payload (CHAMP) (Wickert et al., 2004; Wickert et al.,
2001b); Gravity Recovery and Climate Experiment (GRACE) also Gravity Recovery and Climate
Experiment Follow-on (GRACE-FO) (Wickert et al., 2009); Scientific Application Satellite-C/D
(SAC-C/D) (Hajj et al., 2004); TerraSAR-X; TanDEM-X; Constellation Observing System for
Meteorology, Ionosphere, and Climate (COSMIC/COSMIC-2, also known as FORMOSAT-3/FO
RMOSAT-7); the Meteorological Operational satellite Programme-A/B/C (MetOp-A/B/C);
FengYun-3C/D (FY-3C/D) (Sun, 2019); Communications/Navigation Outage Forecasting
System(C/NOFS); Korea Multi-Purpose Satellite-5 (KOMPSAT-5); the Indian Space Research
Organization spacecraft Ocean Satellite-2 (OceanSat-2); and Spanish PAZ (peace in Spanish). A
few missions were retired, such as COSMIC-1, GRACE, CHAMP, and SAC-C/D, and some
missions are completed by the end of 2020, such as FY-3C, TanDEM-X/TerraSAR-X,
KOMPSAT-5, OceanSat-2, and C/NOFS. More missions are planned for the future like MetOp
Second Generation (MetOp-SG), FengYun-3E/F/G/H (FY-3E/F/G/H), TerraSAR-X Next
Generation (TSX-NG), Jason Continuity of Service-A/B (JASON-CS-A/B", also known as
Sentinel 6A/6B), and Meteor-MP N1/N2. By 2025, the planned missions will provide around
14,700 RO profiles daily (Jin et al., 2013; Oscar, 2020)".
In recent years, monitoring the tropopause has received an increased attention for climate
change studies. Many studies signified the tropopause rise as a result of the troposphere warming
caused by the increase of the GHGs emissions in the atmosphere (Davis and Rosenlof, 2012; Davis
and Birner, 2013; Staten et al., 2018; Grise et al., 2019; Watt-Meyer et al., 2019; Meng et al.,
2021; Pisoft et al., 2021). The tropopause characteristics are important for the understanding of
the exchange of troposphere-stratosphere (Holton et al., 1995). In addition, the chemical,
dynamical, and radiative connections between the troposphere and stratosphere are crucial to
understand and predict climate change worldwide. Exchanges of water, mass, and gases between
the troposphere and stratosphere occurs through the tropopause. Several studies have investigated
the tropopause over the tropics using different data types, and have revealed the problem of the
TEL shift (Ao and Hajj, 2013; Tegtmeier et al., 2020; Kedzierski et al., 2020). GNSS-RO provided
high accuracy remote sensing observations of the thermal structure of the tropopause and was used
to investigate the trend and variability of the tropopause (Son et al., 2011). Among the most
outstanding advantages of GNSS-RO are their high accuracy of 0.2–0.5 K in estimating
temperature in the UTLS region and vertical resolution of 200 m. These advantages make GNSS-
RO especially appropriate to detect the possible tropical belt widening based on the height metrics
of the tropopause (Kursinski et al., 1997; Ho et al., 2012). Using tropopause metrics for TEL
determination have many advantages because it can be accurately estimated from remotely sensed
temperature profiles with sufficient vertical resolution, such as GNSS-RO profiles (Davis and
Birner, 2013; Seidel and Randel, 2006).
In this study, we investigate the TEL variability from different sources, mainly GNSS-RO data. In
Section 2, a description of the GNSS-RO data and other data sets used in our study is presented.
In addition, the methods to derive TEL and the data analysis are presented. Section 3 describes
and discusses the results of the analysis, and our conclusion and summary are given in Section 4.

## 143 2. Data and Methods

### 144 2.1. Data

In this study we employ the following data sets:
• The main data used in this study is GNSS-RO atmospheric profiles data from 12 LEO
missions from June 2001 to November 2020. The data (CDAAC, 2021) is available at the
COSMIC Data Analysis and Archive Center (CDAAC). The GNSS-RO data availability
and its time span are shown in Figure 1.

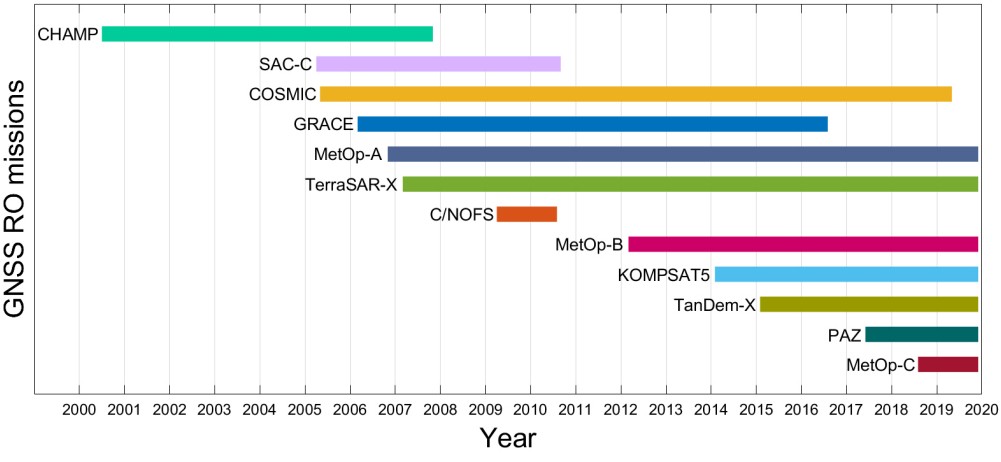

158 **Fig. 1** GNSS-RO data used in this study.

- ERA5 is the fifth generation ECMWF reanalysis for the global climate and weather. Monthly averaged temperature data on pressure levels from ERA5 that provides global coverage for the period from Jun.2001 to Nov.2020 are used to calculate the LRT tropopause height and temperature. The horizontal resolution of the ERA5 data is $0.25^\circ \times 0.25^\circ$, while the vertical coverage covers from 1000 hPa to 1 hPa, with a vertical resolution of 37 pressure levels (Hersbach et al., 2019a).

- The Atmospheric Infrared Sounder (AIRS) is the spectrometer onboard the second Earth Observing System (EOS) polar-orbiting platform, Aqua. In combination with the Advanced Microwave Sounding Unit (AMSU), AIRS constitutes an innovative atmospheric sounding instrument with infrared and microwave sensors. LRT height and temperature data provided by AIRS (AIRX3STM v7.0) are provided monthly and have global coverage, with horizontal resolution of $1^\circ \times 1^\circ$ (Aumann et al., 2003; AIRS, 2019a). In this study we use data for the period from September 2002 to November 2020. The data is available at AIRS (2019a).

- The Modern-Era Retrospective analysis for Research and Applications version 2 (MERRA-2) provides total column ozone (TCO) at a global scale, monthly, and with a spatial resolution of $0.5^\circ \times 0.625^\circ$. In this work, we use data from June 2001 to November 2020. The data is to be compared with the LRT height from GNSS-RO. In addition, TCO can provide information about the tropics behavior and can help in emphasizing the GNSS-RO outputs (GMAO, 2015).

- CarbonTracker is a Carbon dioxide ($CO_2$) measurement and modeling system developed by NOAA Earth System Research Laboratories (ESRL) to keep track of $CO_2$ sources and sinks throughout the world. Monthly column average $CO_2$ data with a global coverage from June 2001 to March 2019 is used in this study (Jacobson et al., 2020). The data has spatial resolution of $2^\circ \times 3^\circ$. Here we use this data to study the behavior and trend of $CO_2$ which is the most important GHG and the largest forcing component in climate change.

- AIRS provides monthly measurements of Methane ($CH_4$) at 24 pressure levels and spatial resolution of $1^\circ \times 1^\circ$ (AIRS, 2019b). We employ data from September 2002 to November 2020. $CH_4$ plays a crucial role in global warming as it is one of the main GHGs that drives long-term climate change.

- Global monthly average surface temperature data from ERA5 reanalysis has horizontal resolution of $0.25^\circ \times 0.25^\circ$ (Hersbach et al., 2019b). In this study we utilize data from June 2001 to November 2020. The purpose of using this data is to study the impacts of the variability in the tropics on the global climatological parameters.

- Monthly average precipitation data is available from the Global Precipitation Climatology Project (GPCP) at horizonal resolution of $2.5^\circ \times 2.5^\circ$ (Adler et al., 2016). We use data from June 2001 to November 2020. The purpose of this data is to investigate the relation between the tropical belt width and the corresponding precipitation pattern.

- Precipitation and Potential Evapotranspiration (PET): Global monthly average precipitation and PET at horizontal resolution of $0.5^\circ \times 0.5^\circ$ are available from the Climatic Research Unit (CRU) Time-Series (TS). This data is employed to compute the SPEI, meteorological drought index. We utilize data from June 2001 to November 2020. The data

is available at Harris et al. (2020). The SPEI drought index was calculated following the
indications of Vicente-Serrano et al. (2010) and Beguería et al. (2013).
*2.2. Methods*
Due to the use of 12 GNSS-RO missions together in our analysis, we compared the different
missions' profiles to investigate the consistency between data from different sources. After that,
the GNSS-RO temperature profiles with a uniform coverage worldwide have been used to
calculate the tropopause height and the tropopause temperature based on both tropopause
definitions LRT and CPT. According to the definition of World Meteorological Organization
(WMO) "The thermal LRT is defined as the lowest level at which the lapse rate decreases to
2°C/km or less, provided also the average lapse rate between this level and all higher levels within
2 km does not exceed 2°C/km" (WMO, 1957). While, the CPT is indicated by the minimum
temperature in a vertical profile of temperature (Holton et al., 1995). Here, in order to avoid
outliers, the tropopause height values of both definitions are limited between 6-20 km. The results
of LRT and CPT are subsequently gridded into 5° x 5°. Finally, the spatial and temporal variability
of all climatic parameters are investigated using the Principal Component Analysis (PCA)
technique (Calabia and Jin, 2016; Calabia and Jin, 2020). This technique provides a new set of
modes that provide the variance through a linear combination of the original variables, based on
Eigen Decomposition. The solution is a couple of matrices containing the eigenvalues and
corresponding eigenvectors of the initial dataset. Each eigenvector is regarded as a map, the
eigenvalues provide the percentage of the contribution to the total variability, and the temporal
coefficients are used to represent the maps at a given epoch. The first PCA mode has the largest
variance, and the following modes represent the next level of variance, which usually are a residual
variability. For this reason, since the variability of the variables used in this study are mainly driven
by the annual variation, we only employ the first PCA component for each case.
The locations of TEL are estimated from monthly zonal average of LRT height derived
from GNSS-RO, ERA5, and AIRS data. The ERA5 and AIRS tropopause parameters are
resampled at the same resolution of GNSS-RO. The zonal average LRT height is spline
interpolated as a function of latitude (Ao and Hajj, 2013), and the TEL is determined at each
hemisphere using two tropopause height metrics. The first method relies on subjective criterion,
according to the first method TEL defined as the latitude at which the LRT height falls 1.5 km
under the tropical average (15°S–15°N) LRT height (Davis and Rosenlof, 2012). The second
method is an objective criterion, in which the TEL is defined as the latitude of maximum LRT
height meridional poleward gradient (Davis and Rosenlof, 2012). Moreover, the rate of expansion
and/or contraction of the tropical belt is estimated from both calculation methods, at each
hemisphere, independently. In addition, the trend and spatial-temporal variability of $CO_2$ and $CH_4$,
as important drivers of global warming, are investigated. Furthermore, the trend and spatial-
temporal pattern of TCO that give information about the tropical belt width are investigated.
Finally, we broadly examine the surface temperature, precipitation, and drought trends as
meteorological parameters which may have a changing behavior as a response to tropics
expansion.

## 3. Results and Analysis

*3.1- Assessment of GNSS-RO Temperature Profiles*

In several previous studies, multiple GNSS-RO missions were utilized together for the purpose of obtaining high spatial resolution. In addition, the assessment of using different GNSS-RO missions together showed high level of consistency (Hajj et al., 2004; Li et al., 2017; Tegtmeier et al., 2020; Xian et al., 2021). In our study, the atmospheric profiles, from all used GNSS-RO missions, are compared together to signify the high level of consistency and compatibility between RO missions available on CDAAC web, also the ability to merge them together in our study as a single dataset. COSMIC mission profiles are used as a fixed member in the intercomparison of all utilized RO missions as it is the most abundant regarding to profiles density and its time span make overlap with all other missions. The results of the conducted intercomparison show high agreement and consistency between profiles of collocated pairs (Fig. 2). Table 1 demonstrates the results of the collocated GNSS profile pairs. The correlation coefficient between the collocated profile pairs ranges from 0.97 to 0.99 and the temperature mean difference ranges from 0.1 to 0.5 K.

**Table.1** Intercomparison of collocated GNSS profile pairs.

| Mission | Correlation coefficient | Mean difference (k) |
|---|---|---|
| (a) COSMIC – CHAMP | 0.99 | 0.5 |
| (b) COSMIC – SAC-C | 0.99 | 0.2 |
| (c) COSMIC – C/NOFS | 0.99 | 0.32 |
| (d) COSMIC – GRACE | 0.99 | 0.1 |
| (e) COSMIC – MetOp-A | 0.99 | 0.28 |
| (f) COSMIC – TerraSAR-X | 0.98 | 0.22 |
| (g) COSMIC – KOMPSAT5 | 0.97 | 0.13 |
| (h) COSMIC – MetOp-B | 0.99 | 0.14 |
| (i) COSMIC – MetOp-C | 0.99 | 0.47 |
| (j) COSMIC – PAZ | 0.98 | 0.33 |
| (k) COSMIC – TanDem-X | 0.99 | 0.47 |

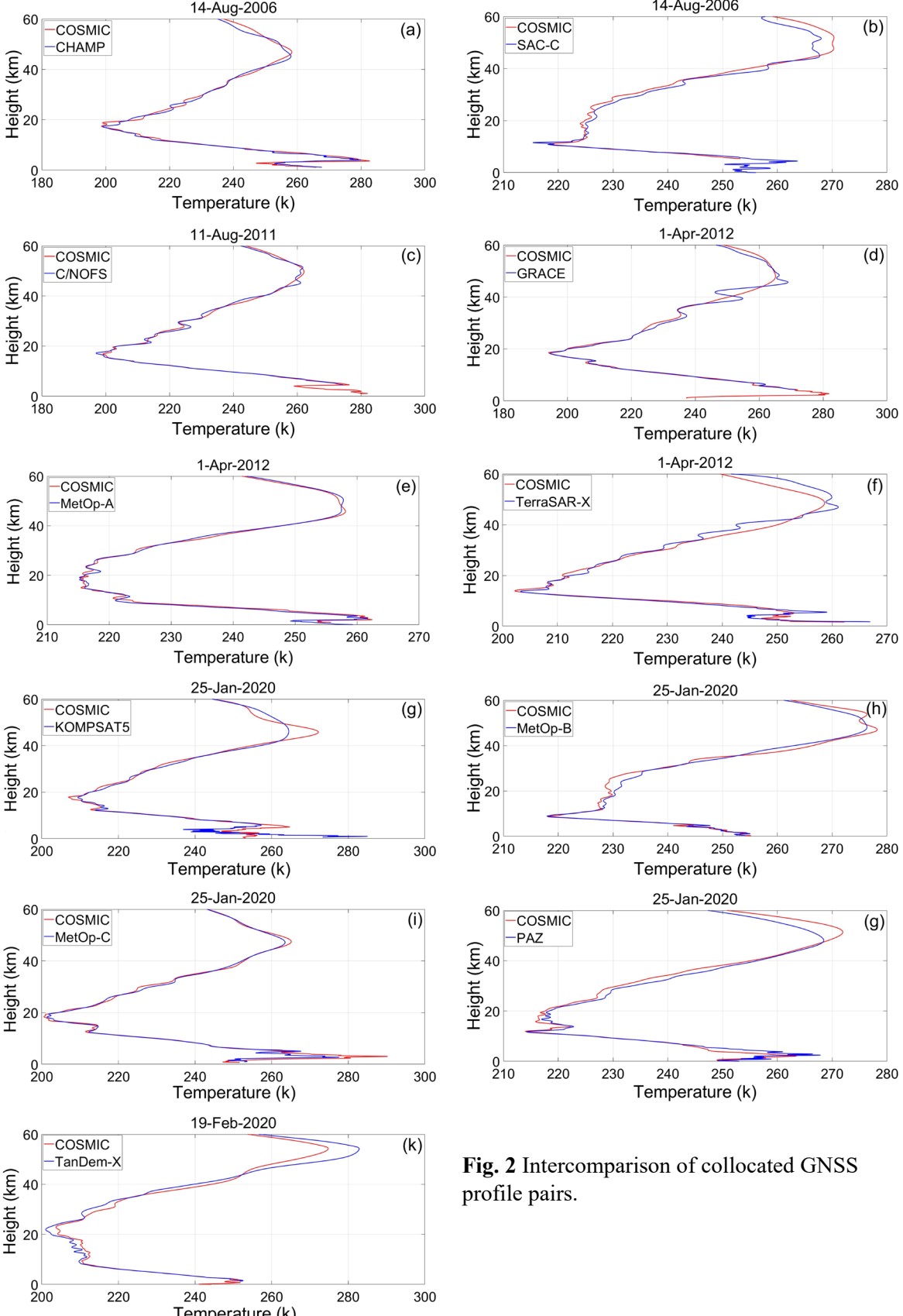

**Fig. 2** Intercomparison of collocated GNSS profile pairs.

*3.2. Tropopause Characteristics from GNSS-RO*

Figure 3 shows the global parameters of GNSS LRT and CPT from June 2001 to November 2020. As clear in Figure 3, the CPT height is always higher than that of LRT. The mean difference between them is about 2.62 km, and there is a correlation of about 0.66 between LRT and CPT height. Previous studies have reported that the average of CPT height is between 0.5 and 1 km higher than the LRT height average (Munchak and Pan, 2014). The LRT temperature is higher than that of the CPT. The mean difference between them is about 4.02 k, and the correlation coefficient between them is 0.61. Our results are consistent with previous studies that displayed a global increase of the tropopause height from radiosonde observations (Seidel and Randel, 2006) and reanalysis (Santer et al., 2004).

Our analysis shows global increasing trend of LRT height of 36 m/decade since 2001 and this has good agreement with that of Schmidt et al. (2008) which shown upward trend of global LRT height of 39–66 m/decade. The LRT temperature show an increase of 0.09 k/decade. For the LRT definition, the correlation coefficient between the LRT height and temperature is -0.78. In case of CPT definition, the global trend of CPT height has increased 60 m/decade since 2001, but that of CPT temperature has decreased 0.09 k/decade. The correlation coefficient between the CPT height and temperature is -0.82.

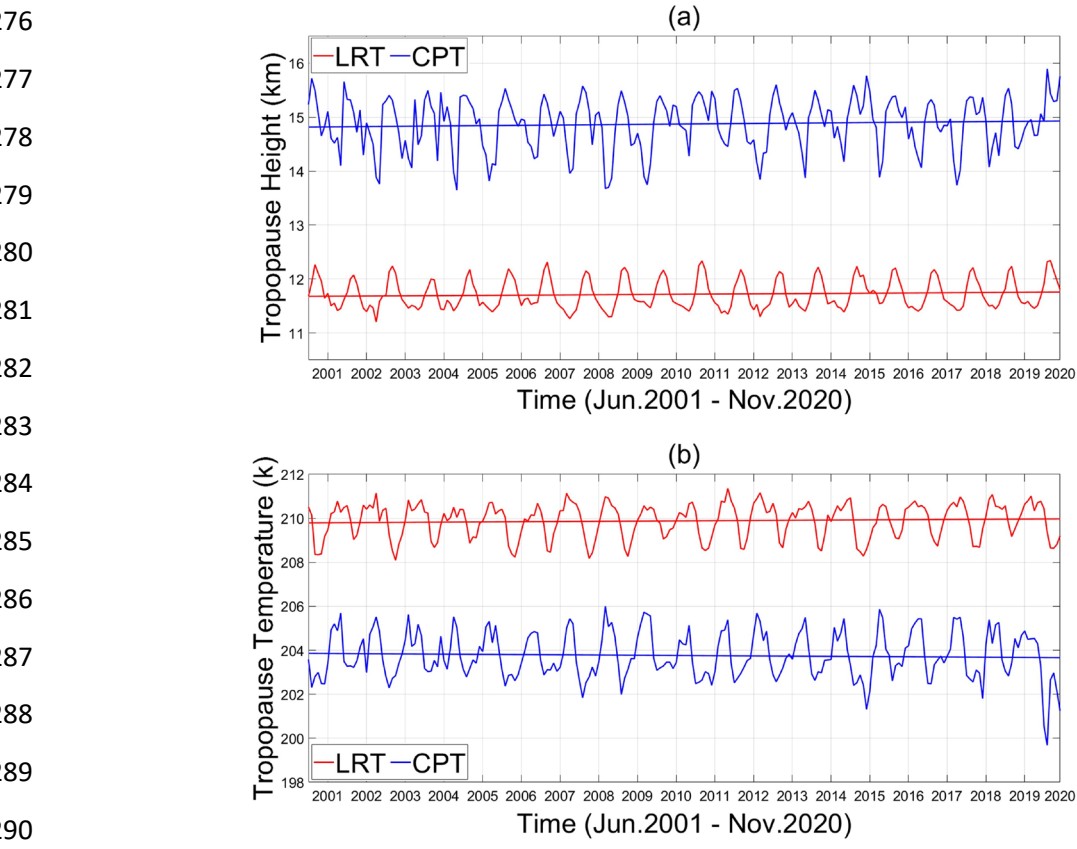

**Fig.3** The LRT and CPT (**a**) height and (**b**) temperature is shown from 2001 to 2020.

*3.3. Comparison between GNSS, ERA5, and AIRS*

In this study, TEL at each hemisphere is estimated from the monthly zonal average tropopause height retrieved from the LRT definition. This is done because the LRT represents the location of the point of thermal transition between troposphere and stratosphere. Furthermore, it reacts to both tropospheric and stratospheric temperature changes. Many studies (Seidel and Randel, 2006; Santer et al., 2004) have shown that LRT height is a good climate change indicator. Figure 4 shows the LRT height values derived from GNSS, ERA5, and AIRS. In general, AIRS shows the highest values of LRT height, while GNSS shows the lowest values. The trends show that ERA5 data has the highest increasing rate of LRT height, being 48 m/decade since June, 2001. In contrast, AIRS has the lowest rates for LRT height, showing an increase of 12 m/decade since September, 2002.

The zonal mean of LRT height for the 3 data sets during January, April, July, and October of 2008 are shown in Figure 5. In January 2008, the high LRT covered higher latitudes in SH than in the NH. The opposite occurs in July. In April 2008, the high LRT covered similarly in both hemispheres. In October, the area covered with high tropopause in NH is larger than that of SH, but not as wide as the coverage in July. This suggest that the warmer the air the wider the area covered with high tropopause. As stated in section 2, the TEL in NH and SH have been estimated applying 2 tropopause height metrics. The results are discussed in detail in the followings.

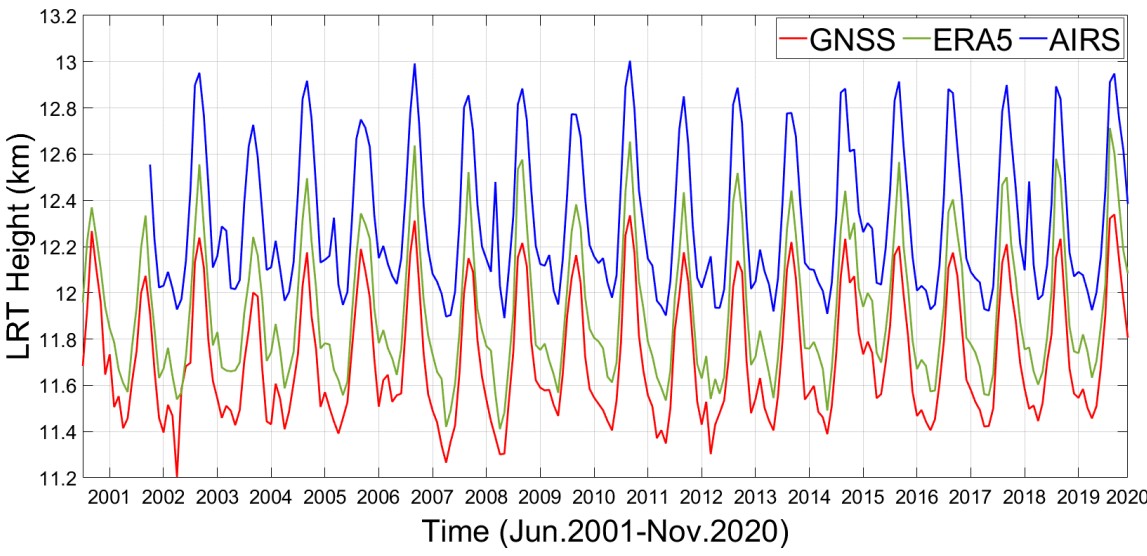

**Fig. 4** LRT height from GNSS, ERA5, and AIRS.

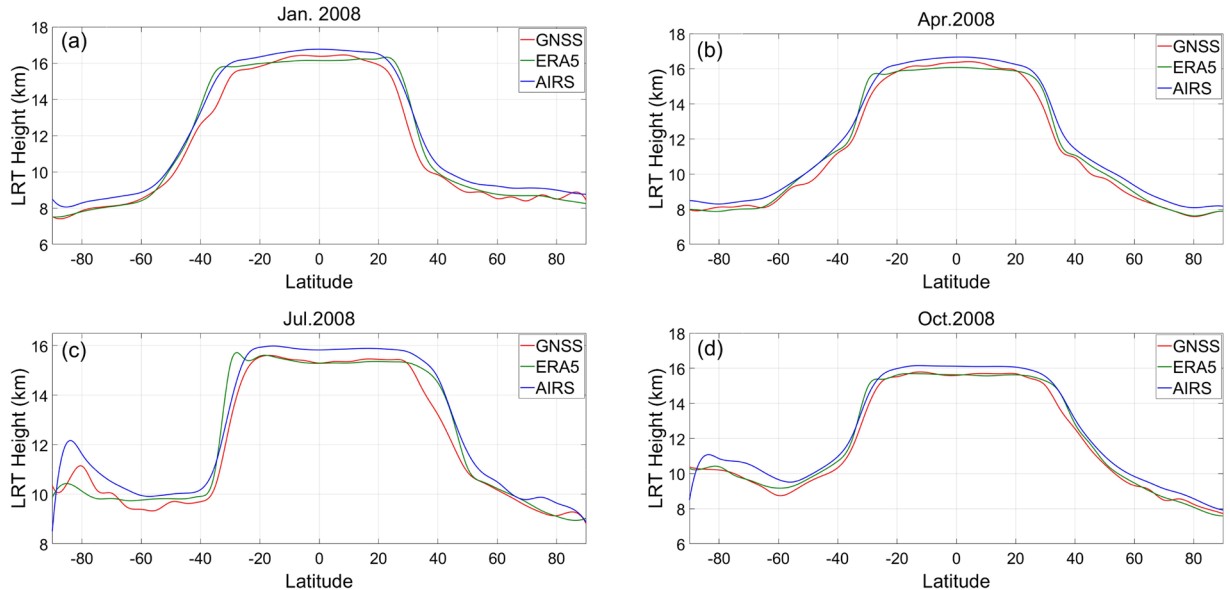

Fig. 5 Monthly zonal average LRT height from GNSS, ERA5 and AIRS.

### 3.3.1. Subjective Criterion for TEL

According to subjective criterion (Davis and Rosenlof, 2012), the TEL at each hemisphere is the latitude at which the tropopause height is 1.5 km under the tropical average tropopause height (15°S-15°N). As shown in Figure 6 and Table 2, the tropical belt based on GNSS has expanded 0.41°/decade in the NH, and 0.08°/decade in the SH, since 2001. Using GNSS-RO data the tropical belt expansion trends in NH and SH agree with the results of Ao and Hajj (2013). According to Meng et al. (2021) the highest trend of LRT height is covering latitudinal band 30°N to 40°N and this possibly caused by the tropical widening and subtropical jet poleward shift over the past four decades (Staten et al., 2018) and this corresponds with our study findings. In case of ERA5, there is no significant expansion or contraction in both hemispheres. While AIRS has expansion of about 0.34°/decade at the NH and strong contraction of about -0.48°/decade at the SH.

Table.2 Tropical belt expansion and contraction rates based on subjective criterion.

| Source | Duration | NH | | SH | |
|---|---|---|---|---|---|
| GNSS | Jun.2001-Nov.2020 | 0.41 | ± 0.09 | 0.08 | ± 0.04 |
| ERA5 | Jun.2001-Nov.2020 | -0.01 | ± 0.1 | -0.04 | ± 0.05 |
| AIRS | Sep.2002-Nov.2020 | 0.34 | ± 0.11 | -0.48 | ± 0.05 |

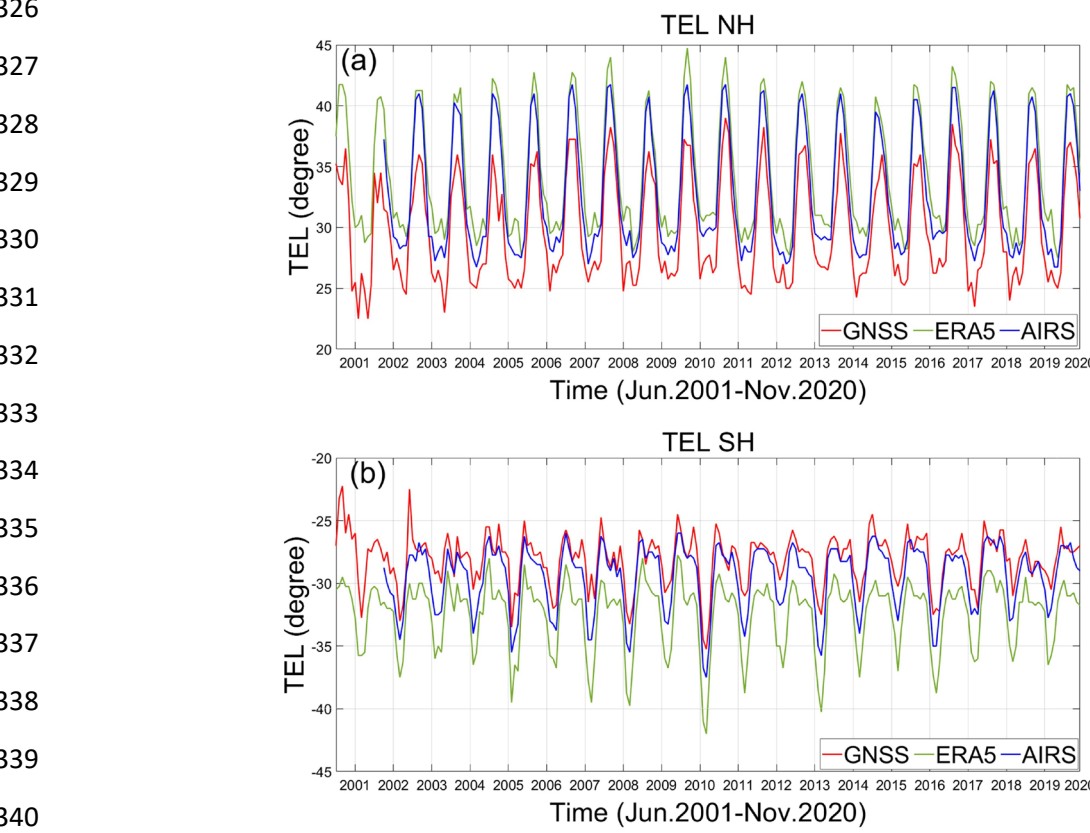

**Fig. 6** TEL using subjective criterion (**a**) NH and (**b**) SH.

### 3.3.2. Objective Criterion for TEL

According to objective criterion (Davis and Rosenlof, 2012), TEL at each hemisphere is the latitude of maximum poleward gradient of tropopause height. As shown in Figure 7 and Table 3, the tropical belt based on GNSS has expanded about 0.13°/decade in the NH since 2001, but there is no significant expansion or contraction in the SH. In case of ERA5, there is no significant trend in NH, while SH has a minor contraction of approximately -0.08°/decade. AIRS has an expansion of 0.13°/decade in NH, and strong contraction in SH of -0.37°/decade. It is clear from these results, that the rates of expansion and contraction using the objective criterion are less than that of the subjective criterion. While in case of the objective method, TEL are located more poleward than that of the subjective method.

**Table.3** Tropical belt expansion and contraction rates based on objective criterion.

| Source | Duration | NH | | SH | |
|--------|----------|------|--------|-------|--------|
| GNSS | Jun.2001-Nov.2020 | 0.13 | ± 0.1 | -0.03 | ± 0.06 |
| ERA5 | Jun.2001-Nov.2020 | -0.06 | ± 0.1 | -0.08 | ± 0.06 |
| AIRS | Sep.2002-Nov.2020 | 0.13 | ± 0.04 | -0.37 | ± 0.06 |

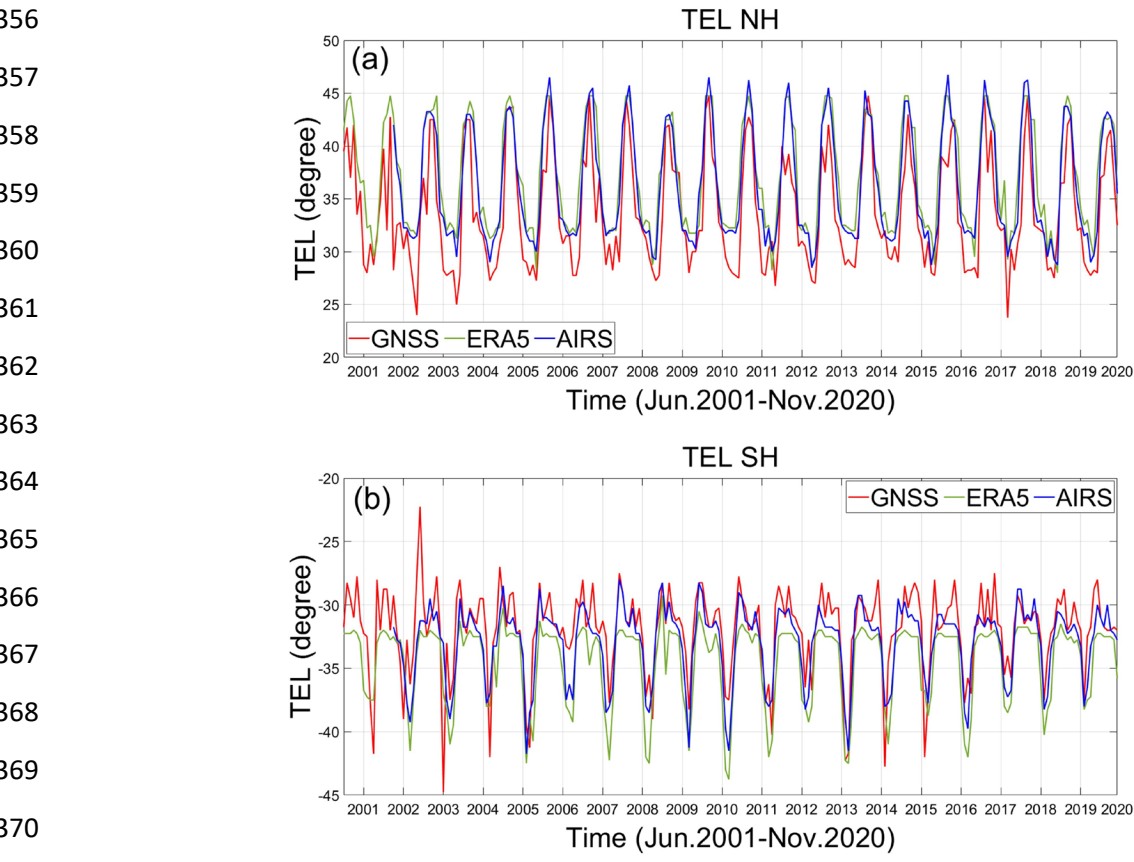

**Fig.7** TEL using objective criterion, (**a**) NH and (**b**) SH.

*3.4. Spatial and Temporal Variability of LRT*

In this section, the GNSS LRT height and temperature between 50°N to 50°S are investigated (Fig. 8). In the NH, the LRT height has increased about 48 m/decade since 2001 and this is consistent with the results of Meng et al. (2021) which shown increase of LRT height around 44.4 m/decade over 20°N to 80°N for the period from 2001 to 2020. In contrast, LRT height in the SH shows a slight decrease of -2.4 m/decade. Regarding to LRT temperature, it has increased about 0.21 k/decade in NH and 0.34 k/decade in SH. Both hemispheres LRT temperature time series show increasing rates higher than the global one 0.09 k/decade. Figure 8 also shows the temporal and spatial variability given by the 1st PCA. The temporal variability for LRT Height captures 22.79% of the total variance. For the LRT temperature, PCA1 captures 13.47% of the total variability. These values are relatively small, showing that the variability spreads along lower degree PCA modes. We can clearly see the annual forcing. The spatial variability shows similar patterns for LRT height and temperature. The signal at the NH is stronger and wider than that at the SH.

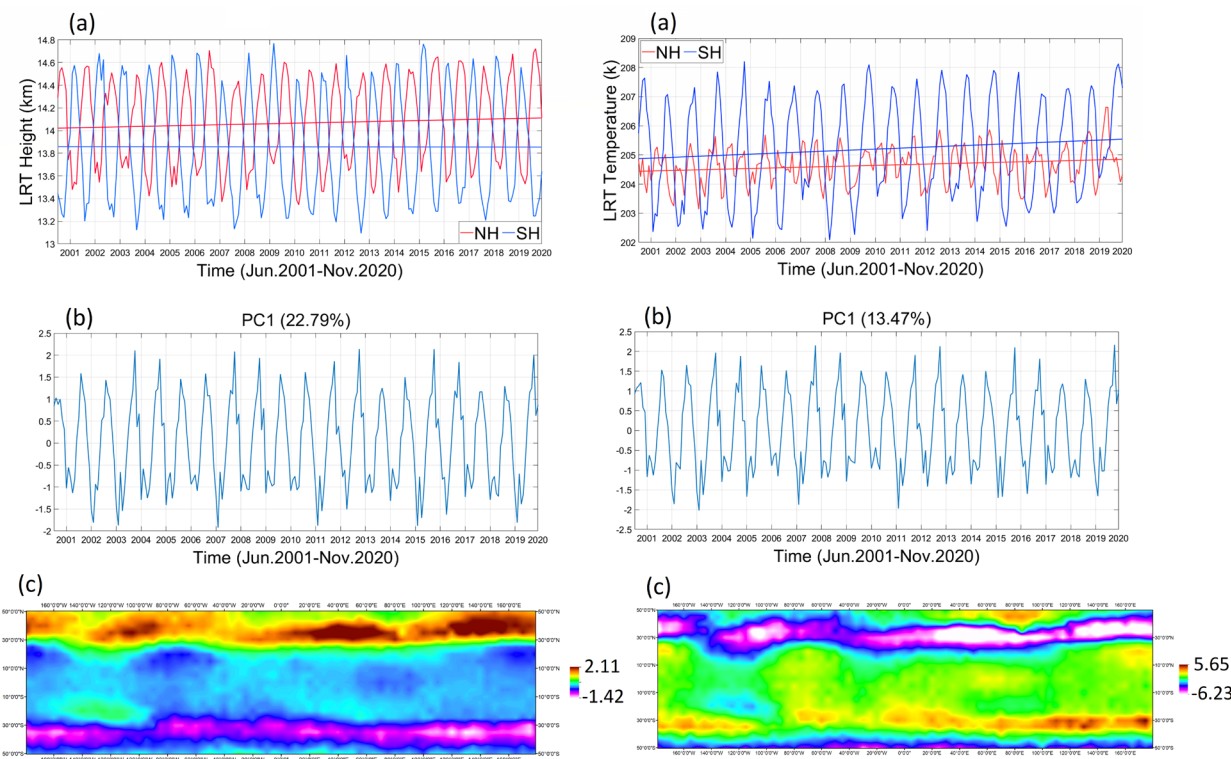

**Fig.8** GNSS-RO based LRT height (left) and temperature (right). In (**a**) temporal time series (**b**) temporal variability given by PCA1, and (**c**) spatial variability map given by PCA1.

### 3.5. Total Column Ozone (TCO), Carbon dioxide (CO₂), and Methane (CH₄)

Figure 9 shows that since 2001, TCO has a global increase of 0.7 DU/decade. TCO has a strong negative correlation of -0.64 with the LRT height. This corresponds with the results of previous work which clarified that TCO pattern is inversely proportion with tropopause height and can give indication about the tropical belt width (Hudson et al., 2003; Hudson et al., 2006; Hudson, 2012; Davis et al., 2018). TCO has increased 0.06 DU/decade and 1.05 DU/decade in the NH and the SH, respectively. Shangguan et al. (2019) reported to asymmetric trends of ozone in the midlatitudes of both hemispheres in the middle stratosphere, with considerable ozone decrease in NH and ozone increase in SH. In our results, the PCA1 of TCO represents 66.68% of the total variability. The spatial map of PCA1 shows stronger signal in the NH than that in the SH. The NH signal is located more poleward than that of the SH. Comparisons with GNSS-RO LRT height spatial and temporal pattern suggest the TCO expansion in the NH, and a weak expansion or non-significant contraction in the SH.

Several studies signified to an increase in the tropopause height as a result of the troposphere warming caused by the rise of the GHGs concentrations in the atmosphere (Meng et al., 2021; Pisoft et al., 2021). $CO_2$ is the most important GHG and it is considered a main driver of global warming. The time series of the $CO_2$ is shown in Figure 9. In this figure, we can see that $CO_2$ has an increase of 21.38 ppm/decade since 2001. It has a correlation of -0.05 with GNSS LRT height. $CO_2$ column average in both NH and SH has the same increasing rate of 21.6 ppm/decade.

This is higher than the global rate. The $CO_2$ standard deviation (STD) in NH is 11.38 which is
higher than that of SH 10.90. The temporal variability given by the PCA1 capture 77.64% of the
total variability. PCA1 shows increasing trend and large variability with time. The map of PCA1
variability shows a shift toward the north pole. This seems to be related to the coverage of the
tropical belt i.e., the TEL occurrence at the NH is more poleward than that of the SH.

412         $CH_4$ is one of the main GHGs, and it is considered a long-term driver of climate change.
The global time series of $CH_4$ column average (Fig. 9a) shows increasing trend of 39 ppb/decade
since 2001. This variable has a correlation of 0.23 with GNSS-RO LRT height. $CH_4$ column
average in both NH and SH show equal increasing trends of 46.8 ppb/decade. This is higher than
the global rate. The $CH_4$ STD in the NH is similar to that in the SH 25.91. The temporal variability
of PCA1 capture 40.65% of the total variability. It shows non-significant trend but its range
increases with time. The map of PCA1 shows more poleward signal in the NH than its equivalent
in the SH. The NH signal reaches 30°N while the SH signal does not reach the limit of 30°S. This
is clearly in with the GNSS TEL results, showing that the tropical condition in the NH covers a
wider area than that in the SH.

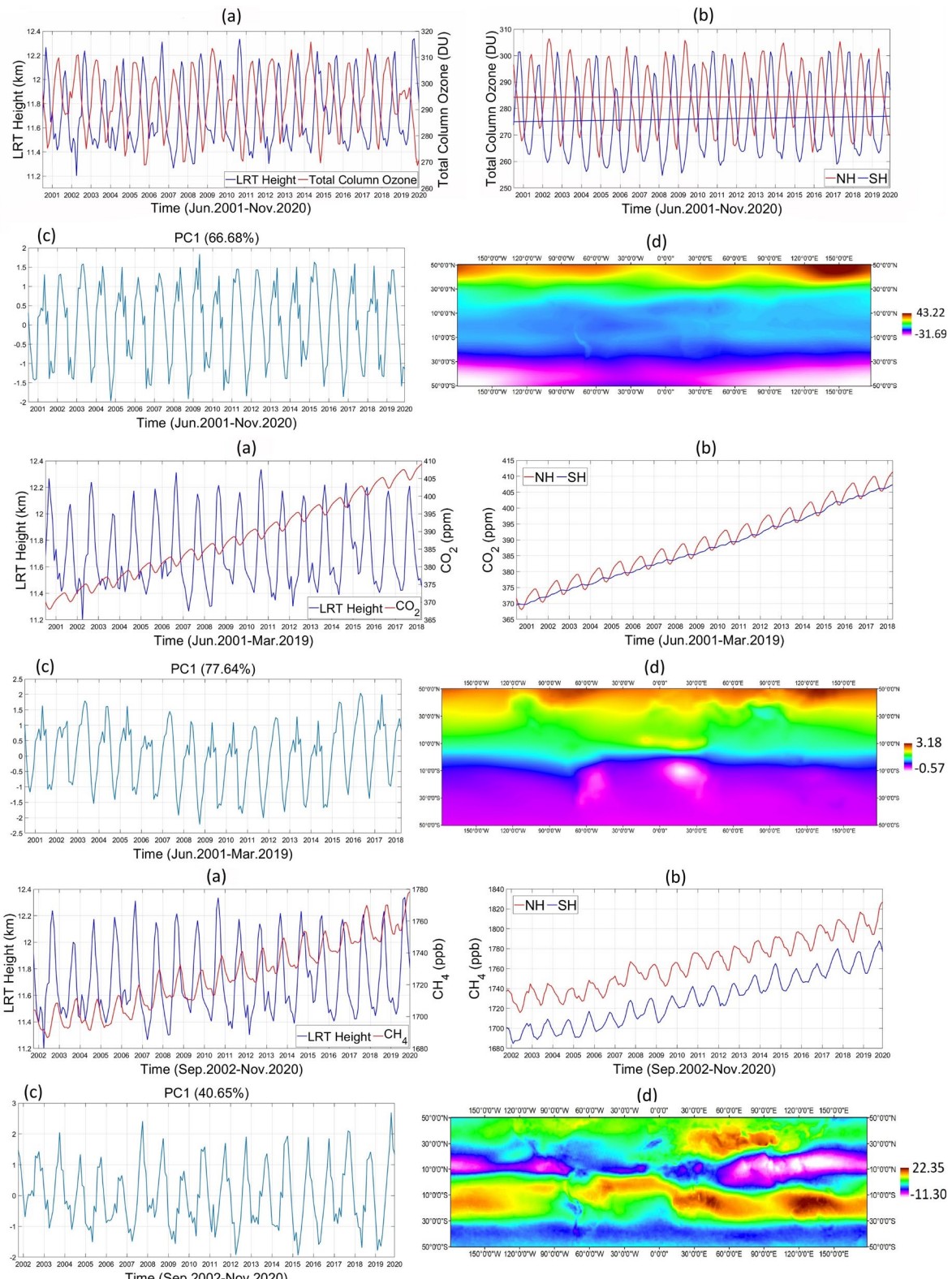

**Fig.9** TCO (top), $CO_2$ (middle), and $CH_4$ (bottom) results. In (**a**) global time series against GNSS LRT height (**b**) temporal time series in NH and SH (**c**) temporal variability given by PCA1 and (**d**) spatial variability map given by PCA1.

*3.6. Surface Temperature and GPCP Precipitation*

Many studies revealed the relation of the surface temperature with the tropopause height and tropical belt expansion. Thuburn and Craig (1997, 2000) found the simulated tropopause height to be sensitive to the surface temperature. Figure 10 shows that the global surface temperature has increased 0.3 k/decade since 2001. A clear correlation between the surface temperature and the GNSS-RO LRT height is seen, with a value of 0.81. The surface temperature in both NH and SH shows increasing trends of 0.23 k/decade and 0.18 k/decade, respectively. The surface temperature in the NH has STD of 3.5 while that of the SH has STD of 1.5. The PCA1 capture 84.41% of the total variance. The PCA1 shows an increasing trend and amplitude with time. The PCA1 map has a signal in SH weaker than that in NH. The results of surface temperature agree with that of GNSS-RO TEL. For instance, the NH show expansive behavior more than that in the SH which shows a minor expansion using subjective criterion and non-significant contraction applying objective criterion. Gao et al. (2015) signified that the correlation coefficient between global tropopause height anomalies and the Niño 3.4 sea surface temperature index is 0.53, with a maximum correlation coefficient of 0.8 at a lag of three months. Fomichev et al. (2007) also found that an increase in sea surface temperature resulted in a tropopause height increase in a coupled chemistry climate model simulation. Hu and Fu (2007) suggested that an increase in sea surface temperatures in the tropics could result in an increase in the tropopause height and a wider Hadley Circulation (tropics width). In addition, our results support surface temperature as a proposed driver for tropics expansion (Allen et al., 2012a; Adam et al., 2014).

The precipitation spatial and temporal variability is investigated to examine the impacts of the TEL variability on the precipitation behavior. The GPCP precipitation has a global decrease of -0.04 mm/decade since 2001. The precipitation behavior has strong correlation of 0.61 with the GNSS LRT height. The GPCP precipitation in the NH show a minor decreasing trend of -0.02 mm/decade meanwhile the SH shows a significant decreasing trend of -1.3 mm/decade. The precipitation in the NH has STD of 15.84 and the SH has STD of 16.47. The PCA1 capture 29.30% of the total variability. PCA1 has upward trend and amplitude with time. The PCA1 map shows a pattern in NH that is stronger and more poleward than that in SH. The precipitation can be used as an independent metric in signifying the TELs locations. Many studies, rely upon surface-based variables to investigate tropical widening, used the GPCP monthly dataset to examine shifts in the positions and boundaries of the subtropical dry zones (Hu et al., 2010; Zhou et al., 2011; Allen et al., 2012b).

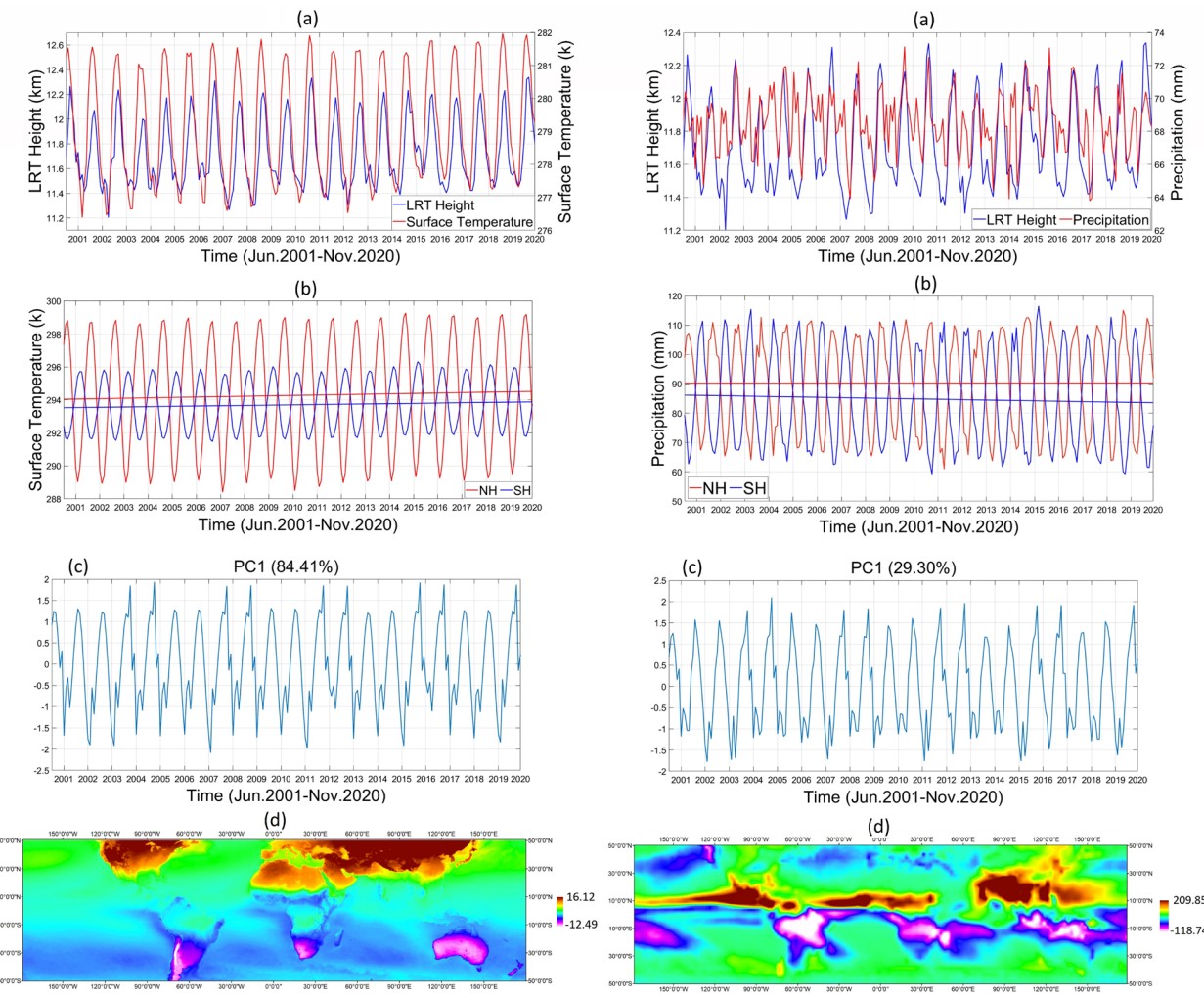

**Fig.10** Surface Temperature (left) and GPCP precipitation (right). In (**a**) global time series against LRT height (**b**) temporal time series in NH and SH (**c**) temporal variability given by PCA1 and (**d**) spatial variability map given by PCA1.

*3.7. Standardized Precipitation Evapotranspiration Index (SPEI)*

The tropical belt widening would contribute in increasing the midlatitude droughts frequency in both hemispheres (Hu and Fu, 2007; Fu et al., 2006; Seidel et al., 2007). The SPEI is usually employed to monitor the meteorological drought status. As clear in Figure 11, the SPEI has a global increase of 0.056 per decade since 2001. The NH shows an increase of 0.035 per decade, and the SH has a decrease of -0.005 per decade. The SPEI has no correlation with GNSS LRT height -0.002. Because the study area is wide and extends through many continents, the SPEI, in our study, only provides information about the dry and wet condition. Figure 11 shows the spatial pattern of SPEI in September 2019, and the areas by category of no-drought, moderate, severe, and extreme. Figure 12 shows the number of cells covered with drought, and its corresponding classification from Figure 11. The total number of cells covered with drought at the NH nearly double its value at the SH. Both hemispheres have a decreasing trend of the number of cells covered with drought. The decrease rate is 510 cell/decade in the NH and 373 cell/decade in the SH. The drought does not show any spatial pattern associated with the locations of TELs.

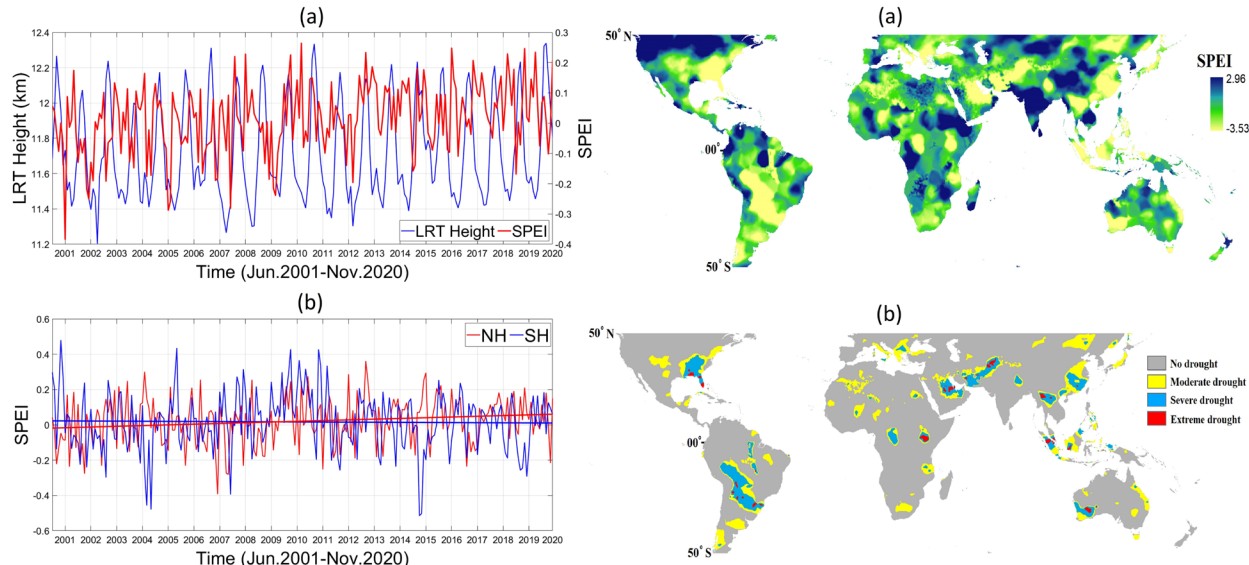

**Fig.11** On the left, SPEI drought index (**a**) global SPEI time series in comparison with LRT height and (**b**) SPEI for two latitudinal bands 0°-50°N & 0°-50°S. On the right, (**a**) SPEI drought index in September 2019 and (**b**) SPEI drought categories in September 2019.

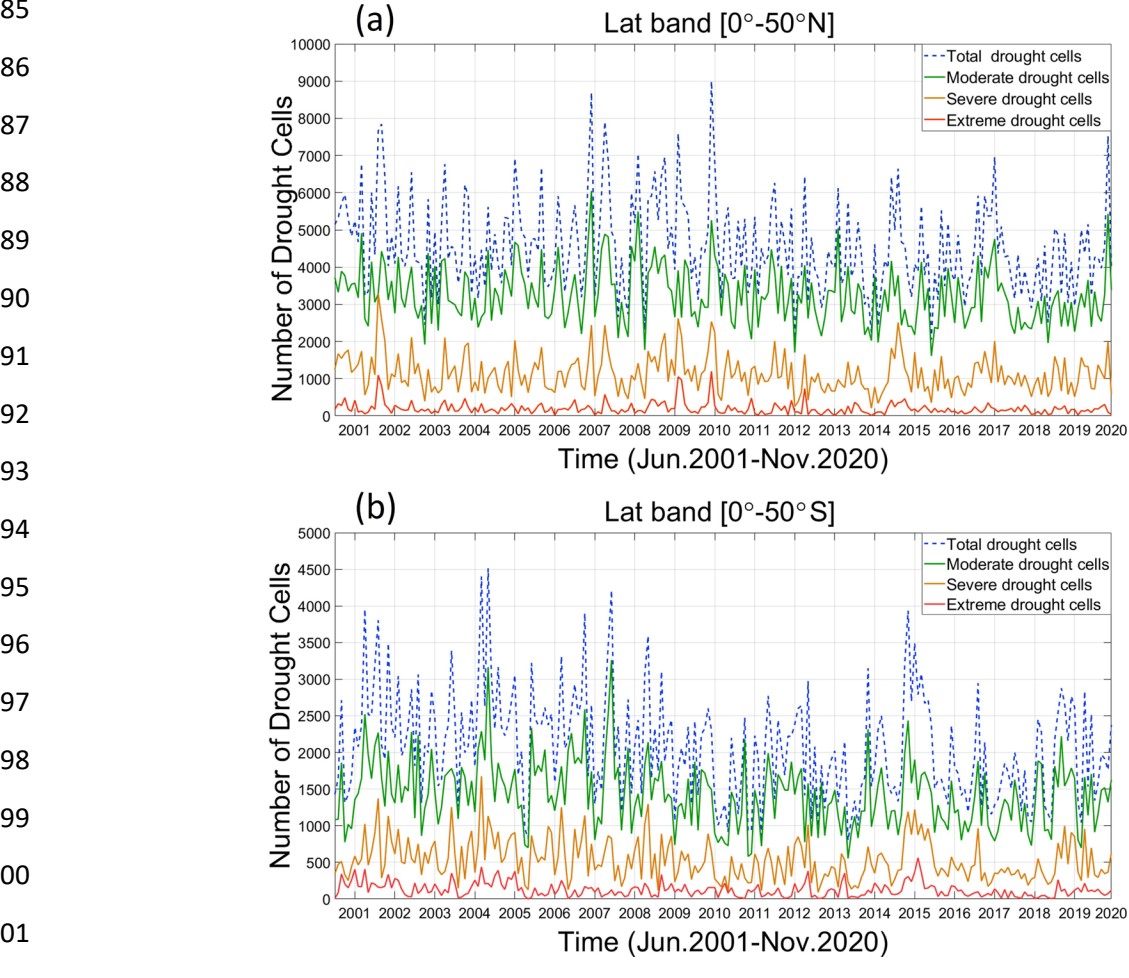

**Fig.12** Number of cells covered with drought at (**a**) NH and (**b**) SH.

## 4. Conclusions

The GNSS-RO is a well-established technique to derive atmospheric temperature structure in the UTLS region. In this study, GNSS-RO data of 12 RO missions are combined together to examine the possible tropical belt expansion. The intercomparison of GNSS-RO profiles of the different utilized RO missions show high level of consistency to be employed together in our analysis. GNSS-RO profiles are employed to derive tropopause height and temperature based on LRT and CPT definitions. The tropopause height is a key element in climate change research because its variability has a correlation with the global warming phenomena (Santer et al., 2003; Sausen and Santer, 2003; Seidel and Randel, 2006; Mohd Zali and Mandeep, 2019). Our analyses show that GNSS LRT and CPT height have increased 36 m/decade and 60 m/decade, respectively, since June, 2001. There is high correlation between the tropopause height and temperature, being -0.78 and -0.82 for LRT and CPT, respectively. The LRT height from ERA5 shows an increase of 48 m/decade since June, 2001 and that derived from AIRS has a smaller increasing rate of 12 m/decade since September, 2002.

In most of the previous studies, the reported tropics widening rates range from 0.25° to 3.0° latitude/decade and their statistical significance vary by large amount based on the metrics used to estimate the TEL as well as the data sets utilized for its derivation (Davis and Rosenlof, 2012). In our study, TEL at each hemisphere is estimated using two tropopause height metrics. Applying the first method, subjective criterion, there are higher expansion and contraction rates than that from the second method, objective criterion. While using the objective criterion, the locations of TELs in both hemispheres are more poleward than that from the subjective criterion. Based on the subjective method, tropical width results from GNSS-RO have an expansive behavior in the NH with about 0.41°/decade, and a minor expansion trend in the SH with 0.08°/decade. ERA5 has non-significant contraction in both hemispheres. In case of the AIRS data, there is a clear expansion behavior in the NH with about 0.34°/decade, and a strong contraction in the SH with about -0.48°/decade. Based on the objective method, GNSS-RO has an expansive behavior in the NH with about 0.13°/decade, but there is no significant expansion or contraction in the SH. For ERA5, there is no significant trend for the TEL results in the NH, while there is a minor contraction of about -0.08°/decade in the SH. The AIRS data show an expansion in the NH with 0.13°/decade, and strong contraction in SH with -0.37°/decade. Results of several studies, based on different data sets and metrics, shown an expansive behavior of tropical belt in NH higher than that of SH and this broadly agree with our GNSS-RO based results (Hu and Fu, 2007; Archer and Caldeira, 2008; Hu et al., 2010; Zhou et al., 2011; Allen et al., 2012b). From all data sets, the TEL is located more poleward in the NH than in the SH. For both subjective and objective methods, the TELs reach the latitudes of 44.75°N and 46.75°N, respectively, at the NH. Meanwhile, at the SH the TELs reach the latitudes of 42°S and 44.75°S for subjective and objective methods, respectively. In both hemispheres, the variability of tropopause parameters (temperature and height) is maximum around the TEL locations.

The TCO shows increasing rates globally. The rate in the SH is higher than that of the NH. The ozone variability agrees well with the spatial and temporal modes of TEL estimated from GNSS-RO LRT height and this supports GNSS-RO TEL estimates over that of ERA5 and AIRS.

In addition, $CO_2$ and $CH_4$, as the main GHGs responsible for global warming, concentrations increase cause a tropopause height rise (Meng et al., 2021; Pisoft et al., 2021). In our analysis, both $CO_2$ and $CH_4$ show a global increasing rate. Their upward trends at the NH and the SH are nearly the same. The patterns of TCO and $CO_2$ display good agreement with the TELs locations in NH and SH. They show more poleward occurrence with time and their variability in NH is higher than that of SH. In addition, $CH_4$ has signal in NH occurs more poleward than that in SH. The surface temperature and precipitation both increase with time, and have strong correlation with LRT height. Both variables show an increasing rate at the NH higher than at the SH. The surface temperature shows strong spatial variability pattern that broadly agrees with the TEL locations from GNSS-RO. The spatial pattern of precipitation shows northward orientation. The SPEI meteorological drought index shows increasing rate globally. It has upward trend in NH while having decreasing trend in SH. Since SPEI is multivariate, it has no direct response to the TEL behavior. In both hemispheres, the number of cells covered with drought decreased since 2001. It can be concluded that the tropical belt widening rates are different from data set to another and from metric to another. In addition, TEL behavior in NH is different from that of SH. Furthermore, the variability of meteorological parameters agrees with GNSS TEL results more than with that of other data sets. The study results signify the importance of monitoring the tropopause and TEL parameters which can accurately indicate the climate variability and climate change globally.

**Funding:** This study was supported by the National Natural Science Foundation of China (NSFC) Project (Grant No. 12073012), National Natural Science Foundation of China-German Science Foundation (NSFC-DFG) Project (Grant No. 41761134092), China Scholarship Council (CSC) and Ministry of Higher Education of the Arab Republic of Egypt.

## Author contributions

M.D. provided the main ideas, developed the methodology, conceived and performed the experiments, and analyzed the results; S.G. provided supervision, mentorship, and funding support; A.C. provided manuscript edition and revision tasks; A.S. helped in manuscript writing and editing.

**Competing interests.** The authors declare that they have no conflict of interest.

## Acknowledgements

The authors thanks to the CDAAC for providing GNSS-RO data, NOAA ESRL for providing CarbonTracker CT2019B, and Copernicus Climate Change Service Information for MERRA-2 and ERA5 data. We are grateful to the Climate Research Unit (CRU), the University of Delaware (UDEL), and Global Precipitation Climatology Project (GPCP) for granting access to datasets. The first author especially thanks to National Research Institute of Astronomy and Geophysics, Egypt, and to Nanjing University of Information Science and Technology, China, for granting the scholarship for pursuing his Ph.D.

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
