# Peer review of "Determination of Tropical Belt Widening Using Multiple GNSS Radio Occultation Measurements"

_Annales Geophysicae, 2021_

## Referee Comment (RC1)

Review of "Determination of Tropical Belt Widening Using Multiple GNSS Radio Occultation Measurements" by Darrag et al.

General
In this paper, the authors presented a study of tropical belt widening using satellite measurements. In the first part, they used two sets of tropopause definitions to examine the widening based on the GNSS radio occultation measurements. They also compared the results with those from AIRS and reanalysis data (ERA5). In the second part, they attempted to explain the widening from different aspects, including surface temperature and precipitation, greenhouse gases, and tropospheric column ozone. The studied issue is interesting and meaningful. The topic is suitable for this journal. The presentation is in good quality.

The following are some comments for the authors to consider when revising this paper.

The authors did a good job in the first part. However, the second part is not well done. The authors may just focus on the first part. If they are also interested in exploring the underlying mechanisms for the tropical belt widening, a much in-depth study is required.

In the first part, the authors only presented the results from GNSS RO satellite, AIRS, and ERA5. There is lack of in-depth discussion on these results. For example, why the results are different, which one is more reliable, what the advantages and limitations for each dataset are, how your results compare with literature…

In the second part, the authors presented time series for a variable of interested, and its PC1 time series and spatial distribution. They also calculated correlation coefficients between tropopause height and that variable. However, it is unclear how these analyses related to the tropical belt widening. The discussion is rather superficial. It is hard to understand what the authors' points. For example, how precipitation is connected to the tropical belt widening?

In section 3.3, the authors stated that LRT temperature shows an increasing trend in both hemisphere during the study period. This is strange for the northern hemisphere where tropopause height shows an increasing trend. As known, if tropopause height increases, tropopause temperature generally decreases.

In section 3.5, the authors stated that the correlation coefficient between the surface temperature and the GNSS RO LRT tropopause height is 0.81. Is this high correlation possible? How is this value calculated? How is this value compared with that in the literature?

Specific
L33, replace "pole ward" to "poleward", the same as for the remaining manuscript.

L37-38, please add a statement on what this means to TEL.

L50, TEL first appears in the text.

L64, it should be in Staten et al. (2018) and Adm et al. (2018). The same format applies for the remaining text.

L70, RS first appears in the text.

L90, LEO first appears here and is defined in L97.

L98 and L113, remove ", and ", respectively.

L218-219, what is this function? Is it an area weighted average?

L275, "figure 4" should be "Figure 4". The same applies for the text throughout. For example, in L318, it should be Figure 6 and Table 2.

L321, "On the other side"? this is used a few times, it may not be a correct expression.

L322, "capture" is not a suitable word to use here.

L408, "Precipitation" should be "precipitation", the same for L410.

L473, L476, L478, "For the subjective method", "for the AIRS data,", "In case of objective method", … These are not good expressions. The authors may use different expressions; for example, "based on the subjective method" is better.

Fig. 7, use different letters for the subplots. Two figures (c) are too small.

---

## Author Comment (AC1)

**Dear Referee_1**

Thank you very much for your valuable comments

We replied the comments one by one in its order: -

The authors did a good job in the first part. However, the second part is not well done. The authors may just focus on the first part. If they are also interested in exploring the underlying mechanisms for the tropical belt widening, a much in-depth study is required.

*Answer: -*

Our study main objective is the determination of tropical belt possible expansion using multiple GNSS Radio Occultation (GNSS-RO) measurements. So, we widely focused on the widening rates and their spatial and temporal patterns as the main point of our research. For the purpose of deep understanding of our work results, we investigated the modes and trends of Carbon dioxide ($CO_2$) and Methane ($CH_4$) as the main greenhouse gases (GHGs) responsible for global warming and climate variability. In addition, we studied the total column ozone (TCO) pattern which inversely proportion with tropopause height and can give indication about the tropical belt width (Hudson et al., 2003; Hudson et al., 2006; Hudson, 2012; Davis et al., 2018). Moreover, surface temperature, precipitation and meteorological drought are investigated as meteorological parameters that their spatial and temporal patterns may change as a result of the expanding or contracting tropics.

In our next papers, we are planning to extend our research about tropical belt to include the mechanisms and drivers of tropical belt width variability. Moreover, the implications of tropics widening and/or contraction on many subtropical regions will be clarified. Finally, the future projection of the tropical edge latitude locations and trends until 2100 will be widely studied.

*Referee comment: -*

In the first part, the authors only presented the results from GNSS RO satellite, AIRS, and ERA5. There is lack of in-depth discussion on these results. For

example, **(a)** why the results are different, **(b)** which one is more reliable, **(c)** what the advantages and limitations for each dataset is, **(d)** how your results compare with literature…

*Answer: -*

**(a)**

[revised manuscript text omitted]

**(d)**

There is no universal definition of the TEL that is applicable across all datasets; each sees the transition zone between tropics and subtropics in a different way. So, the responses of the used metrics show different sensitivities. Regarding to previous studies about the tropical belt widening most of them showed widening rates ranging from 0.25°/decade to 3°/decade. It is not clear to what extent this range reflects inherent differences in differing aspects of the circulation and its drivers, versus the use of different reanalyses, datasets, time periods, and details of tropical edge definitions.

Hudson et al. (2006) analyzed satellite observations of atmospheric ozone concentrations, focusing on the well-known distinction between the tropical regions,

where total column ozone concentration is relatively low, and extratropical regions, where it is higher. Their analysis indicates that the area of the northern hemisphere occupied by the tropical region grew during the 25-year period at a rate of 1°/per decade. Using independent set of satellite-based microwave observations of atmospheric temperature, Fu et al. (2006) inferred tropical-belt widening for the period 1979–2005. Noting that stratospheric cooling and tropospheric warming trends are stronger in the 15–45-degree latitude belts of both hemispheres, they estimate a net widening of the tropical belt of about 2° latitude. A third approach, by Seidel and Randel (2007), based on RS and reanalysis data using tropopause height changes in the subtropics to estimate changes in the width of the tropics. they report an expansion of 5 to 8 degrees latitude during 1979–2005. Using two other types of observations, atmospheric reanalysis and satellite observations of outgoing longwave radiation emitted by the Earth, Hu and Fu (2007) also found a widening of the tropical Hadley circulation system, and estimate its magnitude as 2° to 4.5° latitude during period from 1979 to 2005. Ao and Hajj (2013) used GPS RO data over the period 2002 to 2011 and analyzed it to examine the possible expansion of the tropical belt due to climate change. Analysis showed that a statistically significant widening trend 1°/decade was found in the Northern Hemisphere (NH) while in Southern Hemisphere (SH) no statistically significant trends were found.

The different used metrics for determination of tropical belt width and the results of expansion and/or contraction of most of previous work are discussed in details in Davis and Rosenlof (2012); Lucas et al. (2014); Staten et al. (2018) & Adam et al. (2018).
* * *
*Referee comment: -*

In the second part, **(a)** the authors presented time series for a variable of interested, and its PC1 time series and spatial distribution. They also calculated correlation coefficients between tropopause height and that variable. However, it is unclear how these analyses related to the tropical belt widening. The discussion is rather superficial. **(b)** It is hard to understand what the authors' points. For example, how precipitation is connected to the tropical belt widening?

*Answer: -*

**(a)**

We studied the trends, spatial modes and temporal patterns for many parameters [*Carbon dioxide (CO₂), Methane (CH₄), total column ozone (TCO), surface temperature, precipitation and meteorological drought*]. In addition, we calculated the correlation coefficient between lapse rate tropopause (LRT) height and these meteorological variables and these analyses have a strong relation with the tropical belt widening as the tropical edge latitudes (TELs) are determined based on two tropopause height metrics.

In our analyses the pattern of Carbon dioxide ($CO_2$) and Methane ($CH_4$) as the main greenhouse gases (GHGs) responsible for global warming were investigated as proposed drivers for the tropical edge latitudes (TELs) poleward or equatorward shift. Total column ozone (TCO) was utilized to emphasize the tropopause height results and ensure the TELs locations as it can be used as independent metric for TEL calculation. Moreover, surface temperature, precipitation and meteorological drought were used to clarify the implications of the tropical belt width variability along the study period.

**(b)**

As clear from our paper title, our main interest is the determination of the tropical belt possible expansion using multiple GNSS-RO missions. As we stated before, the precipitation spatial and temporal variability is investigated to examine the impacts of the tropical belt expansion/contraction on the precipitation behavior. is this expansion/contraction associated with a change of the precipitation amount and/or behavior?

Furthermore, the precipitation can be used as an independent metric in signifying the TELs locations. Many studies, rely upon surface-based variables to investigate tropical widening, used the Global Precipitation Climatology Project (GPCP) monthly dataset to examine shifts in the positions and boundaries of the subtropical dry zones (Hu et al., 2010; Zhou et al., 2011; Allen et al., 2012).
* * *
*Referee comment: -*

In section 3.3, the authors stated that LRT temperature shows an increasing trend in both hemisphere during the study period. This is strange for the northern

hemisphere where tropopause height shows an increasing trend. As known, if tropopause height increases, tropopause temperature generally decreases.

*Answer: -*

In section 3.3, for northern hemisphere (NH), both LRT height and temperature have increasing trends along the study period and such results do not contradict with the fact that tropopause height and temperature inversely proportion. The following figures show that there is no conflict in having upward trends for both LRT height and temperature at NH and globally. As clear from the following figures especially that of the whole globe, when the LRT height is high (peak) the LRT temperature is low (trough) and vice versa. The inverse proportion between both can be indicated by the high negative correlation of about -0.78 *[L241-L244]*.

[Figure]

*Referee comment: -*

In section 3.5, the authors stated that the correlation coefficient between the surface temperature and the GNSS RO LRT tropopause height is 0.81. **(a)** Is this high correlation possible? **(b)** How is this value calculated? **(c)** How is this value compared with that in the literature?

*Answer: -*

**(a)**

Yes, this high correlation between surface temperature and the GNSS RO LRT height is possible. The results of surface temperature have good accordance with the LRT height results and, as a result, with TEL patterns. Hence, these findings support surface temperature as a proposed driver for tropics expansion (Allen et al., 2012; Adam et al., 2014).

**(b)**

The correlation coefficient between the surface temperature and the GNSS RO LRT tropopause height is calculated using global monthly average time series of both variables.

**(c)**

Several studies revealed the relation of the surface temperature with the tropopause height and tropical belt expansion. Thuburn and Craig (1997, 2000) used an atmospheric General Circulation Model (GCM) to examine different theories regarding the key factors determining tropopause height. They found the simulated tropopause height to be sensitive to surface temperature. Gao et al. (2015) signified that the correlation coefficient between global tropopause height anomalies and the Niño 3.4 sea surface temperature index is 0.53, with a maximum correlation coefficient of 0.8 at a lag of three months. Fomichev et al. (2007) also found that an increase in sea surface temperature resulted in a tropopause height increase in a coupled chemistry climate model simulation. Hu and Fu (2006) suggested that an increase in sea surface temperatures in the tropics could result in an increase in the tropopause height and a wider Hadley Circulation (tropics width). In addition, our results support surface temperature as a proposed driver for tropics expansion (Allen et al., 2012; Adam et al., 2014).

**Specific**

L33, replace "pole ward" to "poleward", the same as for the remaining manuscript.

>>Done

L37-38, please add a statement on what this means to TEL.

>>Done

L50, TEL first appears in the text.

>>Done

L64, it should be in Staten et al. (2018) and Adm et al. (2018). The same format applies for the remaining text.

>>Done

L70, RS first appears in the text

>>Done

L90, LEO first appears here and is defined in L97

>>Done

L98 and L113, remove ", and ", respectively.

>>Done

L218-219, what is this function? Is it an area weighted average?

>> The zonal average LRT height was spline interpolated as a function of latitude.

L275, "figure 4" should be "Figure 4". The same applies for the text throughout. For example, in L318, it should be Figure 6 and Table 2

>>Done

L321, "On the other side"? this is used a few times, it may not be a correct expression.

>>Done

L322, "capture" is not a suitable word to use here.

>>Done

L408, "Precipitation" should be "precipitation", the same for L410

>>Done

L473, L476, L478, "For the subjective method", "for the AIRS data,", "In case of objective method", … These are not good expressions. The authors may use different expressions; for example, "based on the subjective method" is better.

>>Done

Fig. 7, use different letters for the subplots. Two figures (c) are too small.

>>The subplots are arranged in two columns; the left is LRT height and the right is LRT temperature. Done.
* * *

---

## Author Comment (AC2)

**Dear Referee_2**

Thank you very much for your valuable comments

We replied the comments one by one in its order: -

**Major comments**

*Referee comment: -*

1) The data of the atmospheric profiles came from very different GNSS-RO sources (Fig 1) and different time of missions. It means that accuracy, data time-rate, region of the atmosphere under RO-sounding very differ from one source (mission) to another. In turn, it may bring uncertainties and mistakes in the long lasting data interpreting. I would recommend to the authors to add correspondent explanation in Section 2 and in Conclusion section.

*Answer: -*

In several previous studies, multiple GNSS-RO missions were utilized together for the purpose of obtaining high spatial resolution. In addition, the assessment of using different GNSS-RO missions together showed high level of consistency (Hajj et al., 2004; Li et al., 2017; Tegtmeier et al., 2020; Xian et al., 2021).

*We added clarification and explanation about the validity of using multi GNSS-RO missions together in our paper methodology, results and conclusion sections.*

"In our study, the atmospheric profiles, from all used GNSS-RO missions, are compared to signify the high level of consistency and compatibility between RO missions available on the COSMIC Data Analysis and Archive Center (CDAAC) web, also the ability to merge them together in our study as a single dataset. COSMIC mission profiles are used as a fixed member in the intercomparison of all utilized RO missions as it is the most abundant regarding to profiles density and its time span make overlap with all other missions. Although the compared profiles are collocated within 3h time spacing and 230 km spatial spacing, the results of the conducted intercomparison show high agreement and consistency between profiles of collocated pairs. The following table and figure demonstrate the results of the collocated GNSS profile pairs. The correlation coefficient ranges from 0.97 to 0.99 and the mean of differences of temperature values between the collocated profile pairs ranges from 0.1 to 0.5 K."

[Figure]

**Fig.** Intercomparison of collocated GNSS profile pairs.

**Table.** Results of the intercomparison of collocated GNSS profile pairs.

| Mission | Correlation coefficient | Mean difference (k) |
|---|---|---|
| (a) COSMIC – CHAMP | 0.99 | 0.5 |
| (b) COSMIC – SAC-C | 0.99 | 0.2 |
| (c) COSMIC – C/NOFS | 0.99 | 0.32 |
| (d) COSMIC – GRACE | 0.99 | 0.1 |
| (e) COSMIC – MetOp-A | 0.99 | 0.28 |
| (f) COSMIC – TerraSAR-X | 0.98 | 0.22 |
| (g) COSMIC – KOMPSAT5 | 0.97 | 0.13 |
| (h) COSMIC – MetOp-B | 0.99 | 0.14 |
| (i) COSMIC – MetOp-C | 0.99 | 0.47 |
| (j) COSMIC – PAZ | 0.98 | 0.33 |
| (k) COSMIC – TanDem-X | 0.99 | 0.47 |

*Referee comment: -*

2) In my opinion, there is luck of discussion of Fig 2-10. There is only list of facts with no even minimal comments. I suppose that minimal discussion for each figure is necessary, something like this: the results on Fig correspond (or contradict) to the physical model of the process (or the known results [Reference 1, Reference 2 et al.]). It can be explained by….. et al.

*Answer: -*

>>Done

We added more comments and detailed discussions for all figures to illustrate the results and draw a good conclusion about our study findings.

*Referee comment: -*

3) In my opinion Conclusion section should consist of more detailed explanation of the unfolded trends in the tropopause height increasing. This is the main results of the manuscript which is important in the global weather forecast.

*Answer: -*

We modified the conclusion to contain more in-depth details about our study results and findings.

**Conclusions:**

The GNSS-RO is a well-established technique to derive atmospheric temperature structure in the UTLS region. In this study, GNSS-RO data of 12 RO missions are combined together to examine the possible tropical belt expansion. The intercomparison of GNSS-RO profiles of the different utilized RO missions show high level of consistency to be employed together in our analysis. GNSS-RO profiles are employed to derive tropopause height and temperature based on LRT and CPT definitions. The tropopause height is a key element in climate change research because its variability has a correlation with the global warming phenomena (Santer et al., 2003; Sausen and Santer, 2003; Seidel and Randel, 2006; Mohd Zali and Mandeep, 2019). Our analyses show that GNSS LRT and CPT height have increased 36 m/decade and 60 m/decade, respectively, since June, 2001. There is high correlation between the tropopause height and temperature, being -0.78 and -0.82 for LRT and CPT, respectively. While the LRT height from ERA5 shows an increase of 48 m/decade since June, 2001 and that derived from AIRS has a smaller increase of 12 m/decade since September, 2002.

In most of the previous studies, the reported tropics widening rates range from 0.25° to 3.0° latitude/decade and their statistical significance vary by large amount based on the metrics used to estimate the TEL as well as the data sets utilized for its derivation (Davis and Rosenlof, 2012). In our study, TEL at each hemisphere is estimated using two tropopause height metrics. Applying the first method, subjective criterion, there are higher expansion and contraction rates than that from the second method, objective criterion. While using the objective criterion, the locations of TEL at both hemispheres are more poleward than that from the subjective criterion. Based on the subjective method, tropical width results from GNSS-RO have an expansive behavior in the NH with about 0.41°/decade, and a minor expansion trend in the SH with 0.08°/decade. ERA5 has non-significant contraction in both hemispheres. In case of the AIRS data, there is a clear expansion behavior in the NH with about 0.34°/decade, and a strong contraction in the SH with about -0.48°/decade. Based on the objective method, GNSS-RO has an expansion behavior in the NH with about 0.13°/decade, but there is no significant expansion or contraction in the SH. Results of several studies, based on different data sets and metrics, shown an expansive behavior of tropical belt in NH higher than that of SH and this broadly agree with our GNSS-RO based results (Hu and Fu, 2007; Archer and Caldeira, 2008; Hu et al., 2010; Zhou et al., 2011; Allen et al., 2012). For ERA5, there is no significant trend for the TEL results in the NH, while there is a minor contraction of about -

0.08°/decade in the SH. The AIRS data show an expansion in the NH with 0.13°/decade, and strong contraction in SH with -0.37°/decade. From all data sets, the TEL is located more poleward in the NH than in the SH. For both subjective and objective methods, the TELs reach the latitudes of 44.75°N and 46.75°N, respectively, at the NH. Meanwhile, at the SH the TELs reach the latitudes of 42°S and 44.75°S for subjective and objective methods, respectively. In both hemispheres, the variability of tropopause parameters (temperature and height) is maximum around the TEL locations.

The TCO shows increasing rates globally. The rate in the SH is higher than that of the NH. The ozone variability agrees well with the spatial and temporal modes of TEL estimated from GNSS-RO LRT height and this supports GNSS-RO TEL estimates over that of ERA5 and AIRS. In addition, CO2 and CH4, as the main GHGs responsible for global warming, concentrations increase cause a tropopause height rise (Meng et al., 2021; Pisoft et al., 2021). In our analysis, both CO2 and CH4 show a global increasing rate. Their upward trends at the NH and the SH are nearly the same. The patterns of TCO and CO2 display good agreement with the TELs locations at NH and SH. They show more poleward occurrence with time and their variability in NH is higher than that of SH. In addition, CH4 has signal at NH occurs more poleward than that at SH. The surface temperature and the precipitation both increase with time, and have strong correlation with LRT height. Both variables show an increasing rate at the NH higher than at the SH. The surface temperature shows strong spatial variability pattern that broadly agrees with the TEL locations from GNSS-RO. The spatial pattern of precipitation shows northward orientation. The SPEI meteorological drought index shows increasing rate globally. The NH shows increasing trend while SH shows decreasing trend. Since SPEI is multivariate, it has no direct response to the TEL behavior. In both hemispheres, the number of cells covered with drought decreased since 2001. It can be concluded that the tropics widening rates are different from data set to another and from metric to another. In addition, TEL behavior in NH is different from that of SH. Furthermore, the variability of meteorological parameters agrees with GNSS TEL results more than with that of other data sets. The study results signify the importance of monitoring the tropopause and TEL parameters which can accurately indicate the climate variability and climate change globally.

**Minor comments**

*Referee comment: -*

1) Abstract: In my opinion Abstract is very long and difficult to catch the main idea of the research. All the numerical evaluations and its short discussion should be in the main text, but not in abstract. Abstract should be short and clear for readers. It should consists of following points: motivation; general list of means of data treatment (or theoretical analysis), experiment environment et al; main results and its novelty declaration comparing to the known results.

*Answer: -*

We modified the abstract to be short clear and more indicative about the study goal, data sets, methods, results and research conclusion.

**Abstract:**

In the last decades, several studies reported the tropics expansion but the rates of expansion are widely different. In this paper, data of 12 global navigation satellite systems radio occultation (GNSS-RO) missions from June 2001 to November 2020 with high resolution were used to investigate the possible widening of the tropical belt along with the probable drivers and impacts in both hemispheres. Applying both lapse rate tropopause (LRT) and cold point tropopause (CPT) definitions, the global tropopause height shows increase of approximately 36 m/decade and 60 m/decade, respectively. The tropical edge latitudes (TELs) are estimated based on two tropopause height metrics, subjective and objective methods. Applying both metrics, the determined TELs using GNSS have expansive behavior in northern hemisphere (NH) while in southern hemisphere (SH) there are no significant trends. In case of ECMWF Reanalysis v5 (ERA5) there are no considerable trends in both hemispheres. For Atmospheric Infrared Sounder (AIRS), there is expansion in NH and observed contraction in SH. The variability of tropopause parameters (temperature and height) is maximum around the TEL locations at both hemispheres. Moreover, the spatial and temporal patterns of total column ozone (TCO) have good agreement with the TELs positions estimated using GNSS LRT height. Carbon dioxide ($CO_2$) and Methane ($CH_4$), the most important greenhouse gases (GHGs) and the main drivers of global warming, have spatial modes in the NH that are located more poleward than that at the SH. Both surface temperature and precipitation have strong correlation with GNSS LRT height. The surface temperature spatial pattern broadly agrees with the GNSS TEL positions. In contrast,

Standardized Precipitation Evapotranspiration Index (SPEI) has no direct connection with the TEL behavior. The results illustrate that the tropics widening rates are different from data set to another and from metric to another. In addition, TEL behavior in NH is different from that of SH. Furthermore, the variability of meteorological parameters agrees with GNSS TEL results more than with that of other data sets.

*Referee comment: -*

2) Line 72 and Line 74: What do these "…reanalyses trends…" and "…different reanalyses…" mean?

*Answer: -*

In our study, the type of the used reanalyses data are atmospheric reanalyses that are generated through the assimilation of the historical atmospheric observational data spanning an extended period, using a single consistent assimilation (analysis) scheme throughout.

Examples for the atmospheric reanalyses datasets:

- ECMWF Reanalysis v5 (ERA5)
- The Modern-Era Retrospective analysis for Research and Applications v2 (MERRA-2)
- National Centers for Environmental Prediction and the National Center for Atmospheric Research (NCEP/NCAR)
- Japanese 25-year ReAnalysis (JRA-25) & the Japanese 55-year Reanalysis (JRA-55)
- ECMWF re-analysis of meteorological observations from September 1957 to August 2002 (ERA-40)

❖ The reanalyses trends are the trends of any geophysical parameter from different reanalyses datasets. These trends can be biased to reflect changes in both the quality as well as the quantity of the underlying data (Schmidt et al., 2004; Ao and Hajj, 2013).

*Referee comment: -*

3) Line 236: Please check and correct it: "…is no significant correlation 0.21…".

*Answer: -*

>>Done

We modified it to be ["There is a correlation of about 0.66 between LRT and CPT height"].
* * *
*Referee comment: -*

4) Line 241: "global increasing trend of LRT height 241 of 36 m/decade". Looking at the Fig 2 I see this trend for CPT but not for LPT. Please check it.

*Answer: -*

>>Done

We checked it and found it to be correct:
Our analysis shows global increasing trend of LRT height of 36 m/decade since 2001 and global upward trend of CPT height of 60 m/decade since 2001.
* * *
*Referee comment: -*

5) Line 483: What do you mean here: "there is no significant signal in the…"?

*Answer: -*

We mean that there is no observed trend showing any expansion or contraction for the tropical belt using ERA5 data in the NH.

>>We modified it to be ["For ERA5, there is no significant trend for the TEL results in the NH,"]
* * *
**References**

[revised manuscript text omitted]

---

## Author Response (AR2)

Dear Topical editor, Referee_1 and Referee_2

Thank you very much for your valuable comments

This is the Author's point-by-point response to the reviews including relevant changes:

1- Regarding to the minor correction: L22, In Abstract, change "at both hemispheres" to "in both hemispheres". We made the required correction. In addition, all similar cases [at NH, at SH and at both hemispheres] along the manuscript are corrected to be [in NH, in SH and in both hemispheres].